# Time Series Counterfactual Inference with Hidden Confounders

## Abstract

We present augmented counterfactual ordinary differential equations (ACODEs), a new approach to counterfactual inference on time series data with a focus on healthcare applications. ACODEs model interventions in continuous time with differential equations, augmented by auxiliary confounding variables to reduce inference bias. Experiments on tumor growth simulation and sepsis patient treatment response show that ACODEs outperform other methods like counterfactual Gaussian processes, recurrent marginal structural networks, and time series deconfounders in the accuracy of counterfactual inference. The learned auxiliary variables also reveal new insights into causal interventions and hidden confounders.

## 1 Introduction

Decision makers want to know how to produce desired outcomes and act accordingly, which requires causal understanding of cause and effect. In this paper, we consider applications in healthcare, where time series data on past features and outcomes are now widely available. Causality in time series have been long studied in statistics (Box et al., 2008), and allows more powerful analysis than methods on time-independent data, like instrumental variable regression (Stock & Trebbi, 2003). However, temporal causality in statistics and econometrics focuses mainly on passively discovering time lag structure (Eichler, 2012). In contrast, decision-making applications need concrete interventions, which is more amenable to an interventionist approach to causality (Woodward, 2005; Pearl, 2009). To give one example, electronic health records (EHR) in healthcare provide an accessible history of a patient's disease progression over time, together with their treatment records and their results. To identify effective treatments, a doctor may want to ask counterfactual questions (Johansson et al., 2016), like "Would this patient have lower blood sugar had she received a different medication?" Through such counterfactual analysis, medical professionals may hope to discover new cures and improve existing treatments. Similar situations arise in other use cases. For example, a user interface designer may want to ask "Would the user have clicked on this ad had it been in a different color?", substantiating their answer from counterfactual inference on clickstream data or other user behaviors.

Counterfactual inference in time series has studied, assuming that all possible causal variables are observed (Soleimani et al., 2017; Schulam & Saria, 2017; Lim, 2018). In practice, however, this assumption of perfect observability is not testable and too strong for many real-world scenarios (Bica et al., 2020). For example, there are many ways in general to treat cancer, but each patient requires their own bespoke treatment plan based on unique characteristics of each case such as drug resistance and toxic response (Vlachostergios & Faltas, 2018; Kroschinsky et al., 2017; Bica et al., 2020). However, these factors are also likely to be unmeasurable in practice, or otherwise not recorded in EHRs. Detecting these hidden confounding variables is therefore crucial to avoid bias in the estimation of treatment effects.

The challenge introduced by confounders in counterfactual inference was first studied in the static setting. Wang & Blei (2019) developed a two-step method that estimates confounders with latent factor models, then infers potential outcomes with bias adjustment. However, confounders in time series can have their own dynamics, and can themselves be affected by the history of interventions. Subsequently, Bica et al. (2020) introducing recurrent neural networks (RNNs) into the factor model to estimate the dynamics of confounders. However, this method only works in discrete time setting with a fixed time step, due to how RNNs are structured. In this paper, we consider the continuous-time setting, which is more flexible in practice and provides more insights of the underlying mechanisms

(Chen et al., 2018; Rubanova et al., 2019). The continuous-time setting is particularly important for healthcare, where there are many time-varying treatments, irregularly-sampled or partially observed time series (Soleimani et al., 2017).

The classical modeling approach to dynamics uses ordinary differential equations (ODEs) $d(\boldsymbol{x}(t))/dt = f(\boldsymbol{x}(t))$, encoding domain expertise of underlying mechanisms in the explicit specification of $f$. In contrast, Chen et al. (2018) introduced the concept of neural ODEs by parameterizing $f$ with neural networks, thus allowing dynamics to be described by arbitrarily complicated functions. Several extensions handle even more complicated issues like irregular sampling or switching dynamics (Jia & Benson, 2019; Kidger et al., 2020). However, these methods cannot be directly applied to time series counterfactual inference, as they focus on initial value problems, which cannot describe interventions without explicit modification of $f$ (Kidger et al., 2020). Furthermore, these existing methods can only handle hidden variables by explicitly describing their dynamics and interdependency with interventions, thus limiting their utility when confounders exist.

**Our contributions.** We propose augmented counterfactual ODEs (ACODEs) to predict how a continuous-time time series will evolve under a sequence of interventions. Our method augments the observed time series with additional dimensions to represent confounders. We then construct counterfactual ODEs based on the neural ODE framework to model the effects of incoming interventions. The ACODE model has three key features. First, it allows for the presence of confounders that can reduce the prediction bias. Second, the ACODE can continuously incorporate incoming interventions using neural ODEs and support irregularly-sampled time series. Third, it demonstrates state-of-the-art performance against competitive baselines for counterfactual inference in both simulation of tumor growth and real-world time series of sepsis patients treatment response. Moreover, the ACODE provides an interface between machine learning and dominant modelling paradigm described in differential equations, which allows for well-understood domain knowledge to be applied to time series counterfactual inference. To the best of our knowledge, this represents the first method for counterfactual inference with confounders in the continuous-time setting.

## 2 RELATED WORK

Time series counterfactual inference stems from causal inference (Pearl, 2009; Eichler, 2012). A large body of pioneering work in causal inference focus on causal relations such as structural causal models (Pearl, 2019) and Granger causality (Eichler, 2007). Counterfactual inference, on the other hand, focus on estimating the effects of actionable interventions, which is a pervasive problem in healthcare (Hoover, 2018). In literature, the difference between the counterfactual outcomes if an intervention had been taken or not is defined as the causal effect of the intervention (Pearl, 2009). Originated from the literature on observational studies (Shadish et al., 2002), Rubin's potential outcome framework has been a popular language to formalize counterfactuals and intervention effect estimate (Rubin, 2005; Imbens & Rubin, 2015).

The problem of hidden confounders in counterfactual inference was first studied in the static setting. Wang & Blei (2019) developed theory for adjusting the bias introduced by the presence of hidden confounders in the observational data. They found out that the dependencies in these multiple confounders can be used to infer latent variables and act as substitutes for the hidden confounders. In this paper, we are interested in considering hidden confounders in time series setting which is much more complicated than in the static setting. Not only because the hidden confounders may evolve over time, but also because they might be affected by previous interventions. On the other hand, most existing work on time series counterfactual inference including counterfactual Gaussian processes (CGP) (Schulam & Saria, 2017) and recurrent marginal structural networks (RMSNs) (Lim, 2018) assume there is no hidden confounders, i.e. all variables affecting the intervention plan and the potential outcomes are observed, which is not testable in practice and not true in many cases. Recently, Bica et al. (2020) applied the idea of latent factor models from Wang & Blei (2019) to the deconfounding of time series. However, their proposed method is based on recurrent neural networks, which works only with discrete and regularly-spaced time series.

Differential equations have been introduced into causal and counterfactual inference in previous studies. Rubenstein et al. (2018) showed that equilibrium states of a first-order ODE system can be described with a deterministic structural causal model, even with non-constant interventions. This

line of literature is centering around casual relations, which is a different focus from this work. On the other hand, differential equations with incoming information is a well-studied mathematical problem in the field of rough analysis, which is referred as controlled differential equations or rough differential equations. These approaches directly integrate with respect to incoming processes (Friz & Victoir, 2010; Lyons et al., 2007).

**Neural Ordinary Differential Equations**  Neural ODEs (Chen et al., 2018) are a family of continuous-time models. Starting from an initial state $\boldsymbol{z}(t_0)$, it evolves following a neural network based differential equations. The state at any time $t_i$ is given by integrating an ODE forward in time:

$$\frac{d\boldsymbol{z}(t)}{dt} = f(\boldsymbol{z}(t), t; \theta), \quad \boldsymbol{z}(t_i) = \boldsymbol{z}(t_0) + \int_{t_0}^{t_i} \frac{d\boldsymbol{z}(t)}{dt} dt \tag{1}$$

where $f$ is a neural network parametrized by $\theta$. Given the initial state, states at any desired time stamps can be evaluated with a numerical ODE solver:

$$\boldsymbol{z}_0, \boldsymbol{z}_1, ..., \boldsymbol{z}_N = \text{ODESolve}(f_\theta, \boldsymbol{z}(t_0), (t_0, t_1, ..., t_N)) \tag{2}$$

More importantly, Chen et al. (2018) proposed to use the adjoint method to compute the gradient with respect to the parameters $\theta$ as long as $f$ is uniformly Lipschitz continuous in $\boldsymbol{z}(t)$ and continuous in $t$. This allows ODE solvers to be used as a black box building block in large models.

## 3  PROBLEM FORMULATION

Consider a multivariate time series $\boldsymbol{x}(t)$ and continuous-time time-dependent interventions $\boldsymbol{a}(t)$. The observational data consists of multiple realizations of above mentioned time series and interventions. Given that realizations are independent to each other, we only consider one realization in following part for simplicity. In a realization up to time $t$, we observe $N$ time series data points and their timestamps $\{\boldsymbol{x}_i, t_i\}_{i=1}^N$ along with continuous-time interventions $\{\boldsymbol{a}(s) : s \leq t\}$. We would like to infer the potential outcome under future interventions given all historical information for any potential intervention plan $\{\boldsymbol{a}(s) : s > t\}$. We will abuse the notation of $\boldsymbol{a}_{>t}$ and $\{\boldsymbol{a}(s) : s > t\}$ in following sections. Our goal is to infer the following distribution:

$$p(\boldsymbol{x}(\boldsymbol{a}_{>t}) | \boldsymbol{a}_{\leq t}, \{\boldsymbol{x}_i, t_i^x\}_{i=1}^N) \tag{3}$$

where $\boldsymbol{x}(\boldsymbol{a}_{>t})$ denotes the potential outcome of future time series $\boldsymbol{x}$ under future intervention $\boldsymbol{a}_{>t}$. Although we cannot directly model this objective distribution, we can instead fit a regression model to estimate $p(\boldsymbol{x}_{>t} | \boldsymbol{a}_{>t}, \boldsymbol{a}_{\leq t}, \{\boldsymbol{x}_i, t_i\}_{i=1}^N)$ from observational data (Rubin, 1978). For cases without hidden confounders, this lead to unbiased estimation of potential outcome $p(\boldsymbol{x}(\boldsymbol{a}_{>t}) | \boldsymbol{a}_{\leq t}, \{\boldsymbol{x}_i, t_i\}_{i=1}^N) = p(\boldsymbol{x}_{>t} | \boldsymbol{a}_{>t}, \boldsymbol{a}_{\leq t}, \{\boldsymbol{x}_i, t_i\}_{i=1}^N)$ under certain assumptions, including *sequential strong ignorability* (Fitzmaurice et al., 2008):

$$\boldsymbol{z}(\boldsymbol{a}_{\geq t}) \perp\!\!\!\perp \boldsymbol{a}(t) | \boldsymbol{a}_{<t}, \boldsymbol{x}_{\leq t} \quad \text{for} \quad \forall \boldsymbol{a}_{\geq t} \tag{4}$$

This condition holds if there are no hidden confounders, which cannot be tested in practice since counterfactual outcomes are never observed in practice. With the presence of hidden confounders, the above assumption is no longer valid and

$$p(\boldsymbol{x}(\boldsymbol{a}_{>t}) | \boldsymbol{a}_{\leq t}, \{\boldsymbol{x}_i, t_i\}_{i=1}^N) \neq p(\boldsymbol{x}_{>t} | \boldsymbol{a}_{>t}, \boldsymbol{a}_{\leq t}, \{\boldsymbol{x}_i, t_i\}_{i=1}^N) \tag{5}$$

Consequently, existing methods which infer conditional distribution $p(\boldsymbol{x}_{>t} | \boldsymbol{a}_{>t}, \boldsymbol{a}_{<t}, \{\boldsymbol{x}_i, t_i^x\}_{i=1}^N)$ from observed data would result in biased estimation of potential outcome.

## 4  AUGMENTED COUNTERFACTUAL ORDINARY DIFFERENTIAL EQUATIONS

To address the problem, the key is to reduce inference bias caused by the presence of hidden confounders and capture underlying temporal dynamics and the intervention effects. We propose a two-step method, called augmented counterfactual ordinary differential equations (ACODEs), which first lifts the time series into an augmented space with additional dimensions and then models the augmented time series with neural network parameterized counterfactual differential equations.

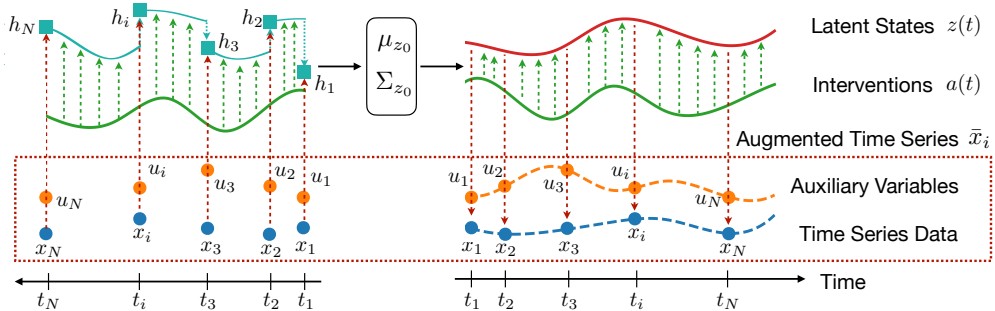

Figure 1: The ACODE model with CDE-RNN encoder and CDE decoder. The CDE-RNN encoder first runs backwards-in-time to produce an approximate posterior over the initial latent state $q(\boldsymbol{z}_0|\{\boldsymbol{x}_i, t_i\}_{i=1}^N, \boldsymbol{a}_{\leq t})$. Given a sample of $\boldsymbol{z}_0$ and intervention process $\boldsymbol{a}(t)$, we can generate latent state at any point of interest, and further generate augmented time series observations.

**Augmented time series** The proposed method first lifts the time series observations by introducing $k$ *auxiliary variables* $\boldsymbol{u}_t \in \mathbb{R}^k$ into $\boldsymbol{x}_t$, resulting in augmented time series $\bar{\boldsymbol{x}}_t = \begin{bmatrix} \boldsymbol{x}_t \\ \boldsymbol{u}_t \end{bmatrix}$. In this augmented space, we can safely assume

$$p(\boldsymbol{x}(\boldsymbol{a}_{>t})|\boldsymbol{a}_{\leq t}, \{\bar{\boldsymbol{x}}_i, t_i\}_{i=1}^N) = p(\boldsymbol{x}_{>t}|\boldsymbol{a}_{>t}, \boldsymbol{a}_{\leq t}, \{\bar{\boldsymbol{x}}_i, t_i\}_{i=1}^N) \tag{6}$$

The insights of introducing *auxiliary variables* $\boldsymbol{u}_t$ come from two aspects. The first one is centered around bias-variance trade off. Basically, modeling in the augmented space can reduce estimation bias at the cost of higher variance (Robins et al., 2000). The second one is about the dimensionality of the space where underlying temporal dynamics work. Basically, we want to approximate the underlying temporal dynamics with learnable mappings. Given the presence of hidden confounders, the true temporal dynamics work in a space with higher dimensionality comparing with the space of time series observations. Therefore, additional dimensions could make it easier to approximate the true temporal dynamics (Dupont et al., 2019).

In general, *auxiliary variables* $\boldsymbol{u}_t$ serve purely as mathematical component without interpretable mechanistic meaning and are initialized with all zero vectors. However, in some cases, the learned *auxiliary variables* $\boldsymbol{u}_t$ can provide interpretable insights about hidden confounders. We will show this later in the experiment on tumor growth simulation. Further, we may also leverage the domain knowledge on hidden confounders and initialize hidden confounders based on their dependency with observed covariates and interventions. This is especially useful in fields where we have mechanistic understanding of hidden confounders thought they cannot be directly measured. For example, a patient's cardiac contractility (the heart's ability to squeeze blood), stroke volume, or systemic vascular resistance are not unobserved, but can be inferred with domain knowledge.

**Latent Counterfactual Differential Equations** Starting from augmented time series, the proposed method models the intervention effects with latent counterfactual differential equations. Specifically, we assume there are latent states $\boldsymbol{z}_t$ representing the state of time series. Latent states evolves controlled by both the baseline progress and intervention effects.

$$\boldsymbol{z}(t) = \boldsymbol{z}(t_0) + \underbrace{\int_{t_0}^t f_z(\boldsymbol{z}(s); \theta_z)ds}_{\text{Baseline progress}} + \underbrace{\int_{t_0}^t f_a(\boldsymbol{z}(s), \boldsymbol{a}(s); \theta_a)ds}_{\text{Intervention effects}} \tag{7}$$

$$\bar{\boldsymbol{x}}(t) \sim p(\bar{\boldsymbol{x}}(t)|\boldsymbol{z}(t)) \tag{8}$$

eq. (7) represents counterfactual differential equations (CDEs), where both $f_z$ and $f_a$ are neural networks parameterized by $\theta_z$ and $\theta_a$ respectively. Unlike previous methods such as (Rubanova et al., 2019), the proposed counterfactual differential equations continuously incorporate incoming interventions, without interrupting the differential equation. Therefore, we can solve the CDEs using the same techniques as for Neural ODEs. Given an initial latent state $\boldsymbol{z}_0$, the generation process for continuous value time series is summarized in Algorithm 1.

---

**Algorithm 1** Generation process of the latent CDE model.

---

**Input**: A distribution of initial latent state $p(\boldsymbol{z}_0)$; timestamps of interest $\{t_i\}_{i=1}^N$; continuous-time intervention process $\boldsymbol{a}_{\leq t}$.
**Output**: Time series observations and their timestamps $\{(\boldsymbol{x}_i, t_i)\}_{i=1}^N$; corresponding latent states $\{(\boldsymbol{z}_i, t_i)\}_{i=1}^N$.
  1: Sample $\boldsymbol{z}_0 \sim p(\boldsymbol{z}_0)$.
  2: Compute $\boldsymbol{z}_1, ..., \boldsymbol{z}_N = \text{ODESlove}(f_z, f_a, \boldsymbol{z}_0, \boldsymbol{a}_{\leq t}, \{t_i\}_{i=1}^N)$
  3: **for** i = 1, ..., N **do**
  4:     Compute $\boldsymbol{\mu}_{x_i}, \boldsymbol{\Sigma}_{x_i} = f_x(\boldsymbol{z}_i; \theta_x)$
  5:     Sample $\boldsymbol{x}_i \sim \mathcal{N}(\boldsymbol{\mu}_{x_i}, \boldsymbol{\Sigma}_{x_i})$
  6: **end for**
  7: **return** $\{\boldsymbol{x}_i, \boldsymbol{z}_i\}_{i=1}^N$

---

**Algorithm 2** The CDE-RNN encoder for general cases.

---

**Input**: Time series observations and their timestamps $\{(\boldsymbol{x}_i, t_i)\}_{i=1}^N$, the continuous-time interventions process $\boldsymbol{a}_{\leq t}$.
**Output**: Hidden states and their timestamps $\{(\boldsymbol{h}_i, t_i)\}_{i=1}^N$
  1: Set $\boldsymbol{h}_0 = \boldsymbol{0}$
  2: **for** i = 1, ..., N **do**
  3:     Update $\boldsymbol{h}_i' = \text{ODESlove}(g_h, g_a, (t_{i-1}, t_i), \boldsymbol{h}_{i-1}, \{\boldsymbol{a}(s) : t_{i-1} \leq s < t_i\})$
  4:     Update $\boldsymbol{h}_i = \text{RNNCell}(h_i', \boldsymbol{x}_i)$
  5: **end for**
  6: **return** $\{\boldsymbol{h}_i\}_{i=1}^N$

---

We use variational autoencoder framework for model training and counterfactual inference. This requires estimating the approximate posterior $q(\boldsymbol{z}_0 | \{\boldsymbol{x}_i, t_i\}_{i=0}^N, \boldsymbol{a}_{\leq t})$. Inspired by Rubanova et al. (2019), we use RNN together with CDE and propose the CDE-RNN model to incorporate time series observations $\{\boldsymbol{x}_i, t_i\}_{i=0}^N$ during encoding. The proposed CDE-RNN, summarized in Algorithm 2, would be an effective way to handle irregularly-sampled time series. To get the approximate posterior of initial latent state $\boldsymbol{z}_0$ at time point $t_0$, we run the CDE-RNN encoder backwrads-in-time from $t_N$ to $t_0$. Then we represent the approximate posterior with Gaussian random variables depending on the final hidden state of an CDE-RNN:

$$q\big(\boldsymbol{z}_0 | \{\boldsymbol{x}_i, t_i\}_{i=1}^N, \boldsymbol{a}_{\leq t}\big) = \mathcal{N}\big(\boldsymbol{\mu}_{z_0}, \boldsymbol{\Sigma}_{z_0}\big) \tag{9}$$

$$\text{where} \quad \boldsymbol{\mu}_{z_0}, \boldsymbol{\Sigma}_{z_0} = g_z\big(\text{CDE-RNN}_\phi(\{\boldsymbol{x}_i, t_i\}_{i=1}^N, \boldsymbol{a}_{\leq t})\big) \tag{10}$$

Here $g_z$ is a neural network mapping the final hidden state of the CDE-RNN encoder into the mean and variance of the approximate posterior of $\boldsymbol{z}_0$. Following autoencoders framework, we jointly learn both the CDE-RNN encoder and CDE decoder by maximizing the evidence lower bound (ELBO):

$$\begin{aligned} \mathcal{L}_{\text{ELBO}}(\theta, \phi) = &\mathbb{E}_{\boldsymbol{z}_0 \sim q(\boldsymbol{z}_0 | \{\boldsymbol{x}_i, t_i\}_{i=1}^N, \boldsymbol{a}_{\leq t})} [\log p_\theta(\boldsymbol{x}_1, ..., \boldsymbol{x}_N)] \\ &- D_{\text{KL}}[q\big(\boldsymbol{z}_0 | \{\boldsymbol{x}_i, t_i\}_{i=1}^N, \boldsymbol{a}_{\leq t}\big) || p(\boldsymbol{z}_0)] \end{aligned} \tag{11}$$

Although with incoming interventions, the whole model is still a ODE-based sequence-to-sequence model. Therefore, we use the adjoint-based backpropagation described in Chen et al. (2018) for training. The overall learning procedure of ACODE is summarized in Algorithm 3.

## 5 EXPERIMENTS

We evaluate the proposed method with two experiments, including a realistic tumor growth simulation (Geng et al., 2017) and a real-world large scale dataset of ICU patients with sepsis, which is extracted from MIMIC-III dataset (Johnson et al., 2016). Through experiments, we answer the following questions: (1) How is the performance of the proposed model for time series counterfactual inference in the presence of hidden confounders, compared to existing state-of-the-art methods? (2) How would the number of *auxiliary variables* affect the performance in the presence of hidden confounders? (3) Do the learned *auxiliary variables* provide any insight of hidden confounders?

---

**Algorithm 3** Learning process of ACODE with variational inference.

---

**Input**: A set of time series along with continuous-time intervention process $\mathcal{D}$; initial value of parameter $(\theta, \phi)$.

1: **while** not converged **do**
2:     Choose a time series with timestamps $\{(\boldsymbol{x}_i, t_i)\}_{i=1}^N \in \mathcal{D}$ and continuous-time intervention process $\boldsymbol{a}_{\leq t} \in \mathcal{D}$.
3:     Initialize *auxiliary variables* $\boldsymbol{u}_i$ with all zero vectors for $i = 1, 2, ..., N$.
4:     Augment time series $\bar{\boldsymbol{x}}_i = \begin{bmatrix} \boldsymbol{x}_i \\ \boldsymbol{u}_i \end{bmatrix}$ for $i = 1, 2, ..., N$.
5:     Initialize hidden state $\boldsymbol{h}_N$ with all zero vectors.
6:     **for** i = N-1, ..., 0 **do**
7:         Update $\boldsymbol{h}_i' = \text{ODESlove}(g_h, g_a, (t_i, t_{i+1}), \boldsymbol{h}_{i+1}, \{\boldsymbol{a}(s) : t_i \leq s < t_{i+1}\})$
8:         Update $\boldsymbol{h}_i = \text{RNNCell}(h_i', \bar{\boldsymbol{x}}_i)$
9:     **end for**
10:    Compute $\boldsymbol{\mu}_{\boldsymbol{z}_0}, \boldsymbol{\Sigma}_{\boldsymbol{z}_0} = g_z(\boldsymbol{h}_0; \phi_z)$
11:    Sample $\boldsymbol{z}_0 \sim p(\boldsymbol{z}_0)$.
12:    Compute $\boldsymbol{z}_1, ..., \boldsymbol{z}_N = \text{ODESlove}(f_z, f_a, \boldsymbol{z}_0, \boldsymbol{a}_{\leq t}, \{t_i\}_{i=1}^N)$
13:    **for** i = 1, ..., N **do**
14:        Compute $\boldsymbol{\mu}_{x_i}, \boldsymbol{\Sigma}_{x_i} = f_x(\boldsymbol{z}_i; \theta_x)$
15:        Sample $\bar{\boldsymbol{x}}_i \sim \mathcal{N}(\boldsymbol{\mu}_{x_i}, \boldsymbol{\Sigma}_{x_i})$
16:    **end for**
17:    Compute the gradient of $\mathcal{L}_{\text{ELBO}}(\theta, \phi)$ as shown in Equation (11).
18:    Update $(\theta, \phi)$ with Adam optimizer.
19: **end while**

---

**Baselines** We compared the proposed method with 3 competitive baselines in the time series counterfactual inference task, including Counterfactual Gaussian Process (CGP) (Schulam & Saria, 2017), Recurrent Marginal Structural Network (RMSN) (Lim, 2018) and Time Series Deconfounder with RMSN (TSD-RMSN) (Bica et al., 2020). To demonstrate the effectiveness of augmented space in ACODE, we remove all *auxiliary variables* (k=0) from ACODE for ablation comparisons.

**Performance Criterion** We compute root mean square error (RMSE) and normalized root mean square error (NRMSE) for each time series averaging across inference time horizon. For each experiment setting, we repeat 10 times and compute the standard deviation as a measure of inference variance.

**Implementation Details** For baselines only work with discrete-time interventions, we discretize the continuous-time intervention process with the same timestamps as time series observations $\{(\boldsymbol{x}_i, t_i)\}_{i=1}^N$, i.e. $\{\boldsymbol{a}_i : \boldsymbol{a}_i = \boldsymbol{a}(t_i)\}_{i=1}^N$. For baselines designed for regularly-spaced time series, like RMSN, TSD-RMSN, we use linear interpolation as the bridge to transfer irregularly-sampled time series to its regularly-spaced counterpart, and vice versa. We use Gaussian distribution with diagonal covariance for the distribution of latent state $\boldsymbol{z}$ and time series observation $\boldsymbol{x}$. All neural network mappings are parametrized with 3-layer MLPs ans `ReLU` activation. For all neural network based methods, we use a similar amount of parameters for fair comparison. We randomly split each dataset into the training/validation/test set, and choose hyperparameters e.g. the number of hidden factors of TSD-RNN based on the validation set.

## 5.1 TUMOR GROWTH SIMULATION

To show the effectiveness of the proposed method, we first evaluate it on a simulated environment with fully control on the hidden confounders - the pharmacokinetic–pharmacodynamic (PK-PD) model of tumor growth under the effects of chemotherapy and radiotherapy proposed by Geng et al. (2017). The tumor volume after $t$ days since diagnosis is modeled as follows:

$$V(t) = \Big(1 + \underbrace{\rho \log \Big(\frac{K}{V(t-1)}\Big)}_{\text{Tumor growth}} - \underbrace{\beta_c C(t)}_{\text{Chemotherapy}} - \underbrace{(\alpha_r d(t) + \beta_r d(t)^2)}_{\text{Radiotherapy}} + \underbrace{e_t}_{\text{Noise}}\Big)V(t-1) \qquad (12)$$

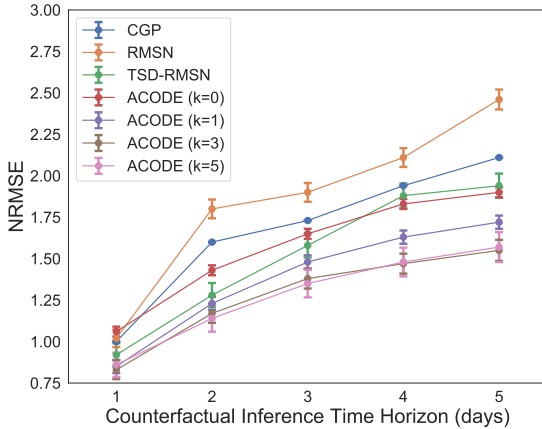

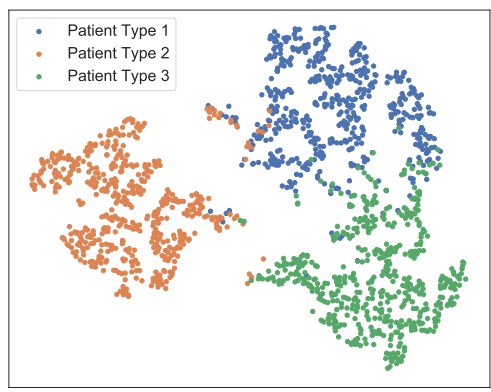

Figure 2: Normalized RMSE curve of counterfactual inference for treatment response on tumor growth.

Figure 3: Visualization of learned *auxiliary variables* sequence $\boldsymbol{u}_t$ for all three types of patients.

where parameter set $\{K, \rho, \beta_c, \alpha_r, \beta_r, e_t\}$ are sampled as described in Geng et al. (2017). Radiotherapy and chemotherapy prescriptions are modeled as Bernoulli random variables which depend on the tumor size $V(t)$. Specifically, we assume the chemotherapy prescriptions and radiotherapy prescriptions have probabilities $p_c(t)$ and $p_d(t)$ respectively that are a functions of the tumor diameter:

$$p_c(t) = \sigma\Big(\frac{\gamma_c}{D_{max}}(\bar{D}(t) - \theta_c)\Big) \qquad p_d(t) = \sigma\Big(\frac{\gamma_d}{D_{max}}(\bar{D}(t) - \theta_d)\Big) \qquad (13)$$

where $\bar{D}(t)$ is the average tumor diameter over the past 15 days, $\sigma(\cdot)$ is the sigmoid activation function, $\theta_c$ and $\theta_d$ are constant parameters, and $\gamma$ controls the degree of time-dependent confounding. In the previous study on the tumor growth (Lim, 2018), the prior means of $\beta_c$ and $\alpha_r$ are adjusted according to three patient types accounting for patient heterogeneity due to genetic features. The patient group corresponds to different parameter setting and thus affects the tumor growth and subsequently the treatment plan. In this experiment, we treat the tumor size as time series observation $\boldsymbol{x}(t)$, patient types as the hidden confounder, treatment plan of chemotherapy and radiotherapy as the intervention $\boldsymbol{a}(t)$. Our task is to predict tumor growth progress under various treatment plans, without any information about patient types.

Following experimental set-up in Lim (2018), we simulated data with 10000 patients for training, 1000 for validation, and 1000 for testing. We set the number of *auxiliary variables* in the augmented space $k \in \{0, 1, 3, 5\}$, and evaluate treatment response inference on tumor size with normalized root mean square errors (NRMSE) and standard deviations.

Figure 2 shows the counterfactual inference accuracy on tumor size across time horizon of 5 days. We observe that, methods considering hidden confounders including the proposed ACODE ($k > 0$) and TSD-RMSN outperform other baselines that assume all confounders are observed such as CGP, RMSN and ACODE ($k = 0$). Further, with the representation power of neural networks and differential equations, ACODE excels all other competitive baselines in term of inference accuracy, especially for long-term inference. Although, *auxiliary variables* in the ACODE can effectively reduce the inference bias caused by hidden confounders, too many *auxiliary variables* may introduce unnecessary variance without much help on inference accuracy. As we can see ACODE ($k = 5$) has similar accuracy comparing with ACODE ($k = 3$), but suffers from a higher variance.

So far we have been treating the *auxiliary variables* $\boldsymbol{u}_t$ as mathematical auxiliary component. Since we force the model to learn the system mechanism in the augmented space $\bar{\boldsymbol{x}}_t = \begin{bmatrix} \boldsymbol{x}_t \\ \boldsymbol{u}_t \end{bmatrix}$, we would like to know whether there is insight learned by *auxiliary variables* $\boldsymbol{u}_t$. Therefore, we randomly choose 1500 patients and project time series of *auxiliary variables* $\{\boldsymbol{u}_i\}_{i=1}^N$ learned by ACODE ($k = 3$) on the two dimensional plane using t-SNE (Maaten & Hinton, 2008), as shown in Figure 3. As we can see, the learned *auxiliary variables* can be clustered into three groups corresponding to three

Table 1: Average RMSE$\times 10^2$ and standard error with 10 runs for the inference of sepsis patients.

| Methods | White blood cell count | Blood pressure | Oxygen saturation |
|---|---|---|---|
| CGP | $2.53 \pm 0.05$ | $9.31 \pm 0.06$ | $1.21 \pm 0.04$ |
| RMSN | $2.91 \pm 0.05$ | $10.29 \pm 0.05$ | $1.74 \pm 0.03$ |
| TSD-RMSN | $2.48 \pm 0.06$ | $9.20 \pm 0.12$ | $1.17 \pm 0.08$ |
| ACODE (k = 0) | $2.57 \pm 0.04$ | $9.42 \pm 0.06$ | $1.27 \pm 0.05$ |
| ACODE (k = 1) | $2.47 \pm 0.05$ | $9.35 \pm 0.07$ | $1.15 \pm 0.05$ |
| ACODE (k = 5) | $\mathbf{2.36 \pm 0.07}$ | $\mathbf{8.94 \pm 0.08}$ | $1.09 \pm 0.06$ |
| ACODE (k = 10) | $2.43 \pm 0.11$ | $9.18 \pm 0.13$ | $\mathbf{1.06 \pm 0.09}$ |

different patient types. This pattern demonstrates the potential of ACODE to provide insights of hidden confounders via learned *auxiliary variables* $\boldsymbol{u}_t$.

## 5.2 INTENSIVE CARE OF PATIENTS WITH SEPSIS

Next, we evaluate the proposed method in a real-world scenario without full understanding of hidden confounders - electronic health records of sepsis patients in ICU with three treatment options: antibiotics, vasopressors, and mechanical ventilator. We use the electronic health records extracted from Medical Information Mart for Intensive Care (MIMIC III) database (Johnson et al., 2016). Follow the same pipeline in Bica et al. (2020), we extracted 25 patient covariates consisting of lab tests and vital signs for each patient. In this experiment, we would like to infer the effects of antibiotics, vasopressors, and mechanical ventilator on three patient covariates: white blood cell count, blood pressure, oxygen saturation. Specifically, we extract patient records up to 50 days from MIMIC III database for training and testing, and infer treatment response in 24 hours. The hidden confounders include comorbidities and lab test that are recorded in MIMIC III database but not used by counterfactual inference models. In fact, there might be other hidden confounders, given that it is a real-world scenario.

As we can see in Table 1, the proposed ACODE model outperforms other competitive baselines. Also, there is a clear performance gap between methods considering hidden confounders like ACODE ($k > 0$) and TSD-RMSN, and methods ignoring them like CGP and RMSN. Among all ACODE baselines, with the increasing amount of *auxiliary variables*, ACODE gets lower RMSE at the cost of higher variance. This pattern is consistent with what we observed in the tumor growth simulation. The proper number of *auxiliary variables* $k^*$ varies for individual applications and need to be tuned as a hyperparameter.

## 6 SUMMARY

We proposed the augmented controlled ordinary differential equations (ACODEs) – a novel neural network based model for time series counterfactual inference with the presence of hidden confounders. The proposed method introduces *auxiliary variables* and lifts time series into an augmented space to reduce the inference bias caused by the hidden confounder. With the representation power of neural networks and differential equations, it can effectively capture underlying temporal dynamics and intervention effects from observational data. Empirically, we show that ACODEs outperform existing methods on inference accuracy in both simulations and real-world applications, and showed its potential to provide insight of hidden confounders via *auxiliary variables*. With the increasing amount of data we collect everyday, ACODE would empower us to answer "what if" questions regarding time series and discover insights of underlying mechanisms behind time series observations and possible interventions. For the future work, it would be worth to explore using the ACODE as the interface between machine learning and dominant modelling paradigm described in differential equations, and incorporate well-understood domain knowledge into time series counterfactual inference.

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
