# OpenReview forum: "Time Series Counterfactual Inference with Hidden Confounders"
_ICLR.cc/2021/Conference — Reject_

### Official Review · AnonReviewer3 · 2020-10-27
**Promising direction, but analysis is lacking**

**Rating:** 5
**Confidence:** 3

**Review:**

##########################################################################
Summary:

In this manuscript, the authors propose a novel way of performing counterfactual inference in time-series in the presence of hidden confounders. For this, they employ neural ODEs as a latent time-series model, which they augment with additional latent variables. They test their approach on synthetic and real-world data and demonstrate improved performance in comparison to the state-of-the-art.

##########################################################################
Reasons for score:

Overall, I vote for weak rejection (5) in its current form. This is mainly due to the following reasons:
1.	Marginal theoretical contribution
2.	Analysis of the method is lacking. How critical is the introduction of dynamic variables for the performance? How much is the performance increasing by only increasing the number of parameters of a vanilla neuralODE? How significant are the results, e.g., in figure 2?

##########################################################################
Pros:
1.	The paper is well and concisely written and is understandable to a broad audience.
2.	The authors apply an interesting model to an interesting problem.
3.	To the best of my knowledge, neuralODEs have not been applied to counterfactual inference before, but it is hard to keep track of those two fast-evolving fields (neural ODEs and counterfactuals). I believe, however, that this approach is very promising. The idea of introducing additional dynamic variables to neuralODEs to account for the bias introduced by hidden confounders is also novel.
4.	The presence of clusters in the inferred auxiliary variable states is an exciting finding.

##########################################################################
Cons:
1.	The theoretical contribution is marginal. The authors only slightly extend the model without further studying the implications of the model.

2.	I would have expected to see the following analysis:
A)	Auxiliary states vs. history-dependent (non-Markovian) neuralODE
B)	Auxiliary states vs. additional parameters of neuralODE
This is due to the following reasons: A) In prior work, a similar approach has been employed in various prior works to model hidden confounders, e.g. [Nodelman, U., Shelton, C. R., & Koller, D. (2012). Expectation-Maximization and Complex Duration Distributions for Continuous-Time Bayesian Networks. UAI], or see “Mori-Zwanzig” formalism, in order to model non-Markovian behavior. I would have liked to see some discussion on auxiliary variables vs. non-Markovian neuralODEs (f(x,u,t) vs f(x_1,…x_t,t))?. What is the better approach? To introduce auxiliary variables into the neuralODE, or to extend the neuralODE also to have the time-series’ history as input? B) My kneejerk reaction is that decreased bias and increased variance can be expected when introducing auxiliary variables, solely because this increases the number of parameters of the neuralODE. I would suggest performing a dedicated experiment that allows for disentangling the number of parameters from the presence of auxiliary variables.
3) Further, the results in fig. 2 should include the variance of results. The curves are all close together, and it would be interesting to see whether the improvement is significant.
4) How would I determine k in practice?
##########################################################################
Questions during the rebuttal period:

Please address and clarify the cons above.

#########################################################################
Some typos:
•	Algorithm 1,2,3 “ODESlove”

---

### Official Review · AnonReviewer1 · 2020-10-29
**Paper 3021 Review**

**Rating:** 4
**Confidence:** 4

**Review:**

The authors present an approach to counterfactual inference for time series based on ordinary differential equations to accommodate for the continuous-time setting.

It seems more accurate to refer to the proposed approach as one for counterfactual inference with confounders in the irregularly-spaced time series rather than continuous-time setting.

It may be good for the authors to explicitly point out the similarities and differences of the proposed approach relative to Bica et al, 2020, because on the surface, it seems that the proposed approach simply replaces the RNN in Bica et al, 2020 with the neural ODE model of Chen et al, 2018.

For the first experiment on tumor growth simulation data, for which samples can be obtained at regularly sampled times it is not clear that 1) the linear interpolation for RMSN and TSD-RMSN is necessary (please clarify, because the details of the experiment are quite vague) and 2) assuming samples are regular, why would the proposed approach be better than TSD-RMSN?

The results in Figure 3 are very difficult to evaluate without context about the three groups of patients and whether they will exhibit similar clustering structure should they have clustered from data.

The results in Table 1 are underwhelming considering the error bars of the results relative to the simpler TSD-RMSN. More so, when results for different number of hidden confounders (D_z in Bica et al, 2020) are considered, which they should provided that they are also presented for ACODE for completeness.

---

### Official Review · AnonReviewer2 · 2020-10-29
**Interesting method, but does not deal with the identifiability issue**

**Rating:** 5
**Confidence:** 4

**Review:**

The authors propose a new method, called augmented counterfactual ordinary differential equations (ACODs), to do counterfactual inference on time series data in healthcare. This is done by modelling interventions in continuous time with differential equations augmented by auxiliary confounding variables to reduce bias. They demonstrate the proposed method on tumor growth simulation and sepsis patient treatment response.

Pros:
+ The proposed method can predict how a continuous-time time series will evolve under a sequence of interventions.
+ The proposed method is naturally constructed upon the neural ODE framework by additionally modelling the effects of incoming interventions.

Concerns:
- The major concern in this paper is that the authors did not deal with the identifiability issue of the proposed method for counterfactual inference. It is widely acknowledged that identifiability plays a pivotal role in counterfactual inference, without which any counterfactual quantities cannot be guaranteed. In particular in the methods based on neural networks, it is extremely difficult to address the identifiability issue. In light of this, instead it might be a better idea to provide some counterfactual bounds (Pearl, 2009) for the quantities of interest.
- From the experimental results, it seems that the proposed method does not have an obvious advantage in comparison with the baselines.

---

### Official Review · AnonReviewer4 · 2020-10-30
**A paper working on an important problem, but needs some clarifications**

**Rating:** 5
**Confidence:** 3

**Review:**

This paper proposed to solve an interesting problem: how do we perform counterfactual inference for time series data? The paper follows a study of the problem in the static setting: in the first step, the paper fit an augmented time series $u_t$ as additional confounders, and then perform inference based on the augmented time series. The experiments presented appear promising to the time series inference problem.

Strengths:
+ This is a very important problem that needs careful investigation
+ The application of this method to healthcare data is very important
+ The paper is well written and easy to follow
+ The execution of the experiments

Points for Improvement
I would love to hear some clarification questions from the authors
1. The paper introduces the auxiliary variables $u_t$ and assume they plus $x_t$ satisfy the sequential ignorability. How do we ensure the learned $u_t$ satisfy the condition? I am wondering whether there are metrics (for example, in static case, it would be something similar to a conditional independence test) for the sequential case. And whether this constraint should be reflected in the learning process, for example, adding a regularization function?

2. The paper argues that we should consider the continuous case interventions. However, in the applications/experiments, it appears the interventions and problem settings are all discrete time spaced. For example, the tumor growth function is formulated as V(t) = f * V(t-1), which looks like discrete time points. I would love to hear more discussions on the choice between discrete and continuous, and why we should consider the continuous case.

3. What are the exact number of parameters used by each of the method? I would assume to learn the effect of a continuous time process $a_t$, we need much more parameters, however, the paper said we use similar parameters, so would love to know more the size of the parameters. Maybe the proposed method can utilize more parameters better, while the other methods don't benefit from adding more parameters.

---

### Decision · Program_Chairs · 2021-01-07
**Final Decision**

**Decision:**

Reject

**Comment:**

There are some interesting ideas raised on continuous-time models with latent variables in machine learning. However, the reviewers argue, and I agree, that the connection to causal models as typically required in applications about the effects of interventions is not addressed with as much care as it might have been needed.